# Mechanically Contacted Distributed-Feedback Optical Microcavity

**DOI:** 10.3390/nano12111883

**Published:** 2022-05-31

**Authors:** Yue Liu, Miao Liu, Jingyun Hu, Jiajun Li, Xinping Zhang

**Affiliations:** Institute of Information Photonics Technology, Beijing University of Technology, Beijing 100124, China; liuyue2020@emails.bjut.edu.cn (Y.L.); mliu@emails.bjut.edu.cn (M.L.); hujingyun@emails.bjut.edu.cn (J.H.); ljjjun@emails.bjut.edu.cn (J.L.)

**Keywords:** mechanical contact, distributed feedback, optical microcavity, organic semiconductors, amplified spontaneous emission

## Abstract

We report a construction of distributed-feedback (DFB) optical microcavities, which is realized through mechanical contact between a high-quality planar thin film of a polymeric semiconductor and a large-area homogeneous nanograting. Using poly[(9,9-dioctylfluorenyl-2,7-diyl)-alt-(benzo[2,1,3] thiadiazol-4,8-diyl)] (F8BT) as the active medium for the planar layer, we achieve strong amplified spontaneous emission from such a microcavity with a low threshold. This not only simplifies largely the fabrication techniques for DFB microcavities, but also avoids the unexpected chemical interactions during solution processing between the organic semiconductors and the nanograting materials. Furthermore, high-quality polymer thin films with high surface smoothness and high thickness homogeneity are employed without any modulations for constructing the microcavities. This also suggests new designs of microcavity light-emitting diodes, or even for realizing electrically pumped polymer lasers, simply by metallizing the dielectric nanogratings as the electrodes.

## 1. Introduction

Thin-film organic lasers have been reported extensively using microcavities in various designs of the structures and the fabrication techniques [1,2,3,4,5]. Polyfluorenes have been used as active materials in many investigations on organic semiconductor lasers [6,7,8], where poly[(9,9-dioctylfluorenyl-2,7-diyl)-alt-(benzo[2,1,3] thiadiazol-4,8-diyl)] (F8BT) is one of the typical derivatives for green-emitting devices [9,10]. Distributed feedback (DFB) schemes produced by interference lithography have been the mostly employed designs, where the produced photoresist (PR) structures can be directly used as the DFB structures [11,12,13]. In more cases, the PR structures are transferred into other materials for the consideration of more stable devices, more precisely controllable parameters, or more practical applications. Subsequent micro- or nano-fabrication techniques, including reactive ion-beam etching [14,15], nano-imprinting [16,17], and soft transferring techniques [18,19], have been reported in the construction of the DFB microcavities. Such processes not only make the techniques more complex and more expensive, but also lower the reproducibility and stability of the microcavity structures.

Therefore, a most convenient method is coating the organic semiconductors onto the surface of the dielectric nanogratings. However, some big challenges are encountered during such direct spin-coating process. First, the organic solutions may destroy the template structures if the nanograting materials is dissolvable in the solution. Second, the spin-coated organic layer is strongly modulated by the spatial variation of the structures, resulting in an inhomogeneous active medium and leading to difficulties in the subsequent devices or applications. Thirdly, the grating materials may contaminate the organic semiconductors if they are even slightly dissolvable into the organic solvents. In this work, we report a new design of DFB microcavities using a direct mechanical contact between the independently spin-coated F8BT film and interference-lithography-produced PR grating, intending to solve above challenges for the construction of organic thin-film lasers. Amplified spontaneous emission is achieved with low thresholds and high-quality output beams from such a microcavity, implying simple and promising techniques for realizing DFB microcavities.

## 2. Principles and Preparation of the Microcavity

Figure 1 shows the design and fabrication procedures of the mechanically contacted DFB microcavity. The solution of F8BT in chloroform was first prepared with a concentration of 23 mg/mL. It was then spin-coated onto a fused-silica (FS) substrate at a speed of 1000 rpm for 30 s, so that a high-quality film with a sub-300 nm thickness was produced, as shown in Figure 1a(①) where the FS substrate has a thickness of about 1 mm and a dimension of 15 × 15 mm^2^. Meanwhile, a photoresist (PR) film was produced on a FS substrate by spin-coating the positive photoresist S1805 from Rohm & Haas, as shown in Figure 1a(②) Then, interference lithography was carried out on the finished PR film, where a 360 nm laser was used as the UV laser source. A PR grating was thus produced, however, a thin layer of PR remains underneath the grating structures, as shown in Figure 1a(③). This is always unavoidable, since we intend to achieve a sine-wave shape for the PR grating, and it is difficult to develop into the bottom of the PR film. The two pieces of the FS-based structures are clamped together in a face-to-face manner by two metal clamps, so that the F8BT layer contacts closely on the PR grating, forming a closed microcavity by the DFB grating and F8BT waveguide, where θ_B_ is the Bragg diffraction angle, as shown by the zigzag arrows in white and red in Figure 1a(④).

Figure 1b illustrates the basic principles for the microcavity formed by the PR Bragg-grating and the homogeneous F8BT waveguide, where F8BT, PR, and FS have a refractive index of about n_F_ = 1.68, n_P_ = 1.65, and n_S_ = 1.46, respectively, at a wavelength of about 568 nm. Emission from F8BT under optical excitation propagates along the F8BT waveguide and is diffracted by the Bragg grating, so that the incidence (red arrow) on the F8BT/PR-grating is diffracted (magenta) in an opposite direction. Such a closed feedback loop forms when the second-order Bragg diffraction condition is satisfied as: 2n_F_Λsinθ_B_ = 2λ. The output of the microcavity along the normal of the substrates in two opposite directions are indicated by upward and downward red arrows, which satisfies the grating diffraction condition of n_F_Λ(sinθ_B_ + sin0°) = n_F_Λsinθ_B_ = λ. These output beams can be sent to spectrometers or power meters for performance characterization.

Figure 2a shows the scanning electron microscope (SEM) image of the fabricated PR grating by a sideview of the cross-sectional profile of the structures. The interface between the remaining PR and FS is indicated by a horizontal yellow line, where the remaining PR has a thickness of about 200 nm. We can measure a grating period of 370 nm and a modulation depth of 115 nm using the SEM image in Figure 2a. This result agrees very well with the atomic force microscope (AFM) measurement result in Figure 2b. The microscopic image in Figure 2a,b show high-quality nanograting structures with excellent homogeneity, clear structure edges, and basically sinewave line shapes. This high-quality grating structure supplies good basis for achieving the DFB microcavities through mechanical contacts.

For demonstrating high quality of the F8BT thin film, we measured the SEM image of the cross-sectional profile and the AFM image on the top surface of the F8BT layer, as shown by Figure 2c,d, respectively. Figure 2c shows excellent homogeneity of the F8BT thin film with a thickness of 280 nm. Figure 2d demonstrates the excellent surface smoothness with a roughness of only 3.3 nm. The performance characterized by Figure 2c,d ensures an excellent waveguide by the active medium.

## 3. Spectroscopic Response of the Microcavity Structures

Figure 3a shows the optical extinction spectroscopic response of the PR grating in Figure 1b measured at different angles of incidence (θ_i_ = 0–10°) for TE (left panel) and TM polarizations (right panel). A waveguide resonance mode can be observed between 560 and 570 nm for both polarizations at normal incidence, which split into two branches evolving into opposite directions. These features are basic spectroscopic response for identifying waveguide resonance modes [19,20]. This waveguide resonance mode results from the PR grating with a underneath layer of undeveloped PR, as shown in Figure 2a. At normal incidence, the waveguide resonance mode is degenerated at about 568 nm for TE polarization, as highlighted by a red triangle. In this work, this resonance mode can be used as a good reference for justifying the contact between the F8BT thin film and the PR grating structures, as illustrated in Figure 3b.

The two pieces of the FS-based structures are clamped together in a face-to-face manner by two metal clamps, so that the F8BT layer contacts closely on the PR grating, forming a closed microcavity by the DFB grating and F8BT waveguide, as shown by the zigzag arrows in white and red in Figure 1a(④). As shown in Figure 3b, the strong absorption peaked at about 470 nm corresponds to the main absorption spectrum of F8BT. However, two narrow band spectral features can be observed at 559 and 582 nm, as highlighted by a yellow and magenta triangle, respectively. The one at 559 nm can be identified as the waveguide resonance mode in the PR layer, which can be verified by the plot in the in the inset of Figure 3b, where we include the measurement of the optical extinction spectrum on the pure PR grating shown in Figure 2a by the blue curve. The blue and red spectra agree well for the peaks at about 560 nm, as highlighted by the red and yellow triangles. Thus, we may justify the spectral peak at 582 nm in the inset of Figure 3b as the waveguide resonance mode in the F8BT layer.

The corresponding diffraction condition can be written as:(1)$$n_{F}\Lambda \left(\sin\theta_{i}+\sin\theta_{d}\right)=\lambda$$
where n_F_ is the refractive index of F8BT, θ_i_ and θ_d_ are the incident and diffraction angles inside the F8BT layer, Λ is the grating period, and λ is the wavelength of light. For θ_i_ = 0, we have:(2)$$n_{F}\Lambda \sin\theta_{d}=\lambda$$
where θ_d_ has to satisfy the total reflection condition at the PR/FS interface, or
(3)$$\theta_{d}\geq \sin^{-1}\left(n_{S}/n_{F}\right)$$

Therefore, the appearance of the waveguide resonance mode at 582 nm verifies the true contact of the F8BT with the PR, as well as the formation of the DFB microcavity.

For the convenience of further investigations, we present in Figure 3c the absorption and photoluminescence (PL) spectra of F8BT, which are peaked at 461 and 535 nm, respectively. Clearly, efficient emission of F8BT is distributed within a spectral range from 535 to about 583 nm, which has no overlap with the main absorption spectrum and is the spectral range for the studies on the lasing properties of F8BT. It needs to be noted that the waveguide resonance modes correspond to the destructive interference between the direct transmission and diffraction light beams [20,21]. Therefore, the lasing actions should avoid these two peaks of waveguide resonance mode. Thus, we may anticipate a lasing spectrum at the valley position of the red spectrum in the inset of Figure 3b, which is about 569 nm, as highlighted by a green circle in both Figure 3b,c.

However, for the microcavity mode, the diffraction of the emission light will be different from that of the incident light for the waveguide resonance. The emission from F8BT under optical excitation is diffracted by the grating and totally reflected at the F8BT/FS interface, so that it may propagate along the F8BT waveguide, as highlighted by the white arrows in Figure 1a(④). Such a working condition also holds for the propagation along the opposite direction, as highlighted by the red arrows. This supplies mechanisms for the DFB microcavities. The corresponding condition may be described by a second-order Bragg diffraction process as:(4)$$2n_{\mathrm{eff}}{\Lambda \sin\theta }_{B}=2\lambda_{B}$$
where n_eff_ ≈ n_F_ is the refractive index of the microcavity, Λ is the grating period, θ_B_ is the angle of both the incidence and diffraction light beams for the Bragg grating, and λ_B_ is the wavelength of Bragg resonance. Clearly, the value of θ_B_ does not necessarily equal θ_d_ in (2) for waveguide resonance mode, although the two Equations (2) and (4) are in fact equivalent to each other, supplying a chance for the lasing wavelength not to overlap the waveguide resonance mode. This also explains our discussions on Figure 3b,c by the green-circled spectral position, which is the most probable location of the lasing spectrum.

## 4. Amplified Spontaneous Emission (ASE) Based on the Optical Feedback by the Mechanically Contacted DFB Microcavity

Figure 4a shows a photograph of the experimental setup for investigating the optical feedback in the DFB microcavity and Figure 4b shows a schematic illustration of the optical configuration. Two metal clamps are used to make the two pieces of FS substrates closely contact to each other, where the front piece is coated with a thin layer of F8BT and the back piece is structured with a PR grating, as illustrated in Figure 4a. The two clamps apply a pressure of about 10 N/cm^2^ on the substrates with an area of 15 × 15 mm^2^. The 400 nm laser pulses with a pulse length of 150 fs were used as the pump for investigating the lasing performance of the device, which were produced by frequency-doubling the output of 800 nm pulses from a Ti:sapphire amplifier. The output beam from the mechanically contacted DFB microcavity is shown in Figure 4a, as can be seen by the vertically elongated green spot. However, what we achieved is amplified spontaneous emission (ASE), instead of lasing, which can be justified by the measured output spectrum with a bandwidth of about 11.8 nm at FWHM, as shown in Figure 4c. We plot in Figure 4c the ASE spectra measured at a varied pump fluence from 1.1 to 4.1 μJ/cm^2^, where we can see a clear threshold effect for the spectral feature peaked at 568.3 nm. At a pump fluence of 1.1 μJ/cm^2^, the PL spectrum exhibits two peaks located at 565 and 582 nm, as shown in the inset of Figure 4c, where we plot the PL spectrum of the pure F8BT film by the red curve and that for the clamped microcavity by the black, and the green curve supplies a guideline for identifying the spectral features. The 565 nm peak is the ASE feature at the early stage with the pump fluence just above the threshold and the one at 582 nm can be identified as the weak diffraction and scattering of the waveguide mode in the F8BT layer, which agrees well with the spectral peak in Figure 3b, as highlighted by the magenta triangle. Thus, above mechanisms for the contacted waveguide grating structures are verified again. Furthermore, there is a slight red shift of the ASE peak with increasing the pump fluence.

Figure 4d shows the variation of the peak intensity of the ASE spectrum with the pump fluence. Two stages with distinctly different slopes are observed for the measurement data, which can be well-fitted linearly, convincingly indicating the threshold effect for the ASE emission. We can measure a pump threshold lower than 1 μJ/cm^2^.

According to the experimental results in Figure 4c, we may evaluate a Q value of the microcavity roughly by λ_0_/Δλ ≈ 48.16. This can be used to calculate the cavity loss by $\delta =\frac{2\pi L}{\lambda Q}$, where δ is the single-path cavity loss coefficient, *L* is the effective cavity loss, and *λ* is the wavelength of light. According to our previous publication, the penetration length into the DFB gratings is roughly 10 μm [22], which can be utilized as the effective cavity length. Thus, the single-path loss coefficient can be calculated to be 2.3/cm or 2.3 × 10^−3^/10 μm. This implies a low-loss resonance cavity.

We need to stress that in the attachment process, two metal clamps are applied on the two pieces of glass substrates with a separation equal to the diameter of the effective area of the grating structures. Thus, homogeneous pressure of about 10 N/cm^2^ is applied for the contact between the grating and the F8BT thin film. Since the effect area has a diameter of about 10 mm, we do not expect much mechanical deformation of the clamped glass substrates. Therefore, the production of the DFB microcavity is well reproduced, so that the two substrates can be separated and re-clamped again without any change in the optical performance.

## 5. Conclusions

A DFB microcavity is constructed by mechanical contact between a spin-coated F8BT thin film and a photoresist nanograting, which are fabricated separately without any chemical interactions. This design not only maintains the high quality of the organic layer, but also solves the big challenge of destroying the template grating by the organic solution during spin-coating, as the photoresist may be dissolved in most of the organic solvent. Meanwhile, the contamination of organic semiconductor by other molecules during direct spin-coating onto the template grating is also avoided completely. Optical pumped DFB lasers based on organic thin films can be achieved using such a scheme after optimizing the grating structures and improving the contact quality. This also supplies new approaches for designing organic light emitting diodes or electrically pumped organic lasers, where we simply need to replace the PR grating with a metallic grating as the electrodes.

## Figures and Tables

**Figure 1 nanomaterials-12-01883-f001:**
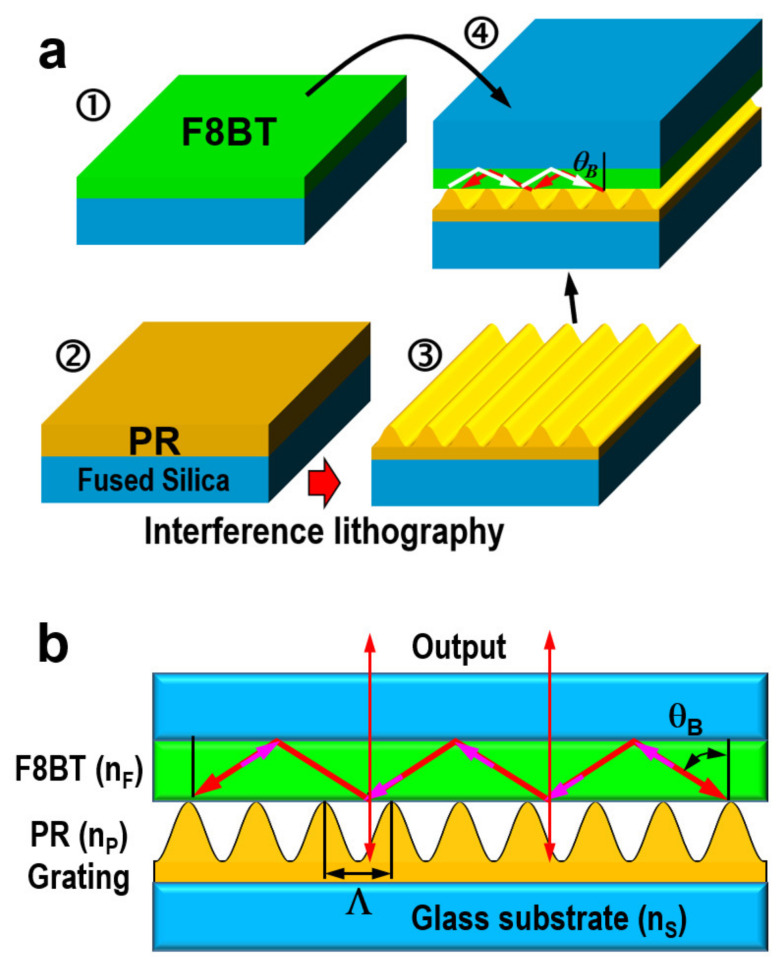
(**a**) Preparation of the mechanically contacted DFB microcavity: (①) spin-coating of a high-quality F8BT thin film on a fused silica substrate; ② spin-coating of photoresist (positive, S1805) onto another piece of fused silica substrate; (③) interference lithography to produce a nanograting into the photoresist; (④) clamp the two finished structures in (①,③) by a face-to-face configuration to accomplish the design of a DFB laser device. (**b**) Schematic illustration of the basic principles for the mechanically contacted microcavity.

**Figure 2 nanomaterials-12-01883-f002:**
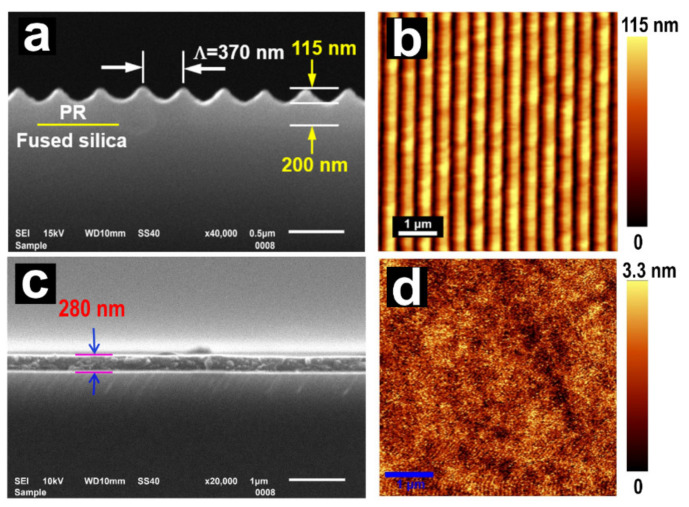
(**a**) SEM image of the cross-sectional profile of the photoresist grating structures. (**b**) AFM image measured on the top surface of the photoresist grating. (**c**) SEM image of the cross-sectional profile of the F8BT thin film, showing a thickness of about 280 nm. (**d**) AFM image measured on the top surface of the F8BT thin film, showing a variation amplitude of only 3.3 nm.

**Figure 3 nanomaterials-12-01883-f003:**
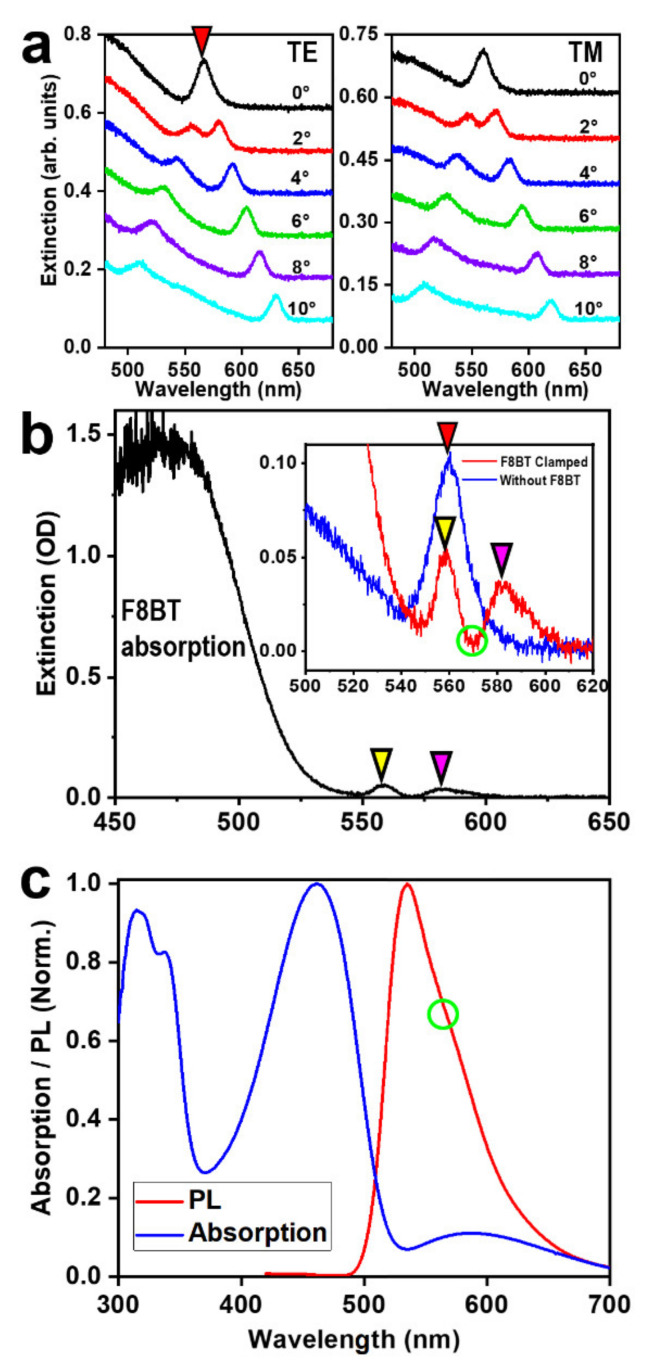
(**a**) Optical extinction spectra measured on the PR grating with a fused silica substrate at different angles of incidence (0~10°) for TE (**left**) and TM (**right**) polarizations. (**b**) Optical extinction measurements on the microcavity formed through clamping the F8BT thin film and the PR grating by a face-to-face scheme, as shown in Figure 1a(④). Inset: enlarged view of the waveguide resonance modes (red) and comparison with that measured on the pure PR grating (blue). (**c**) Absorption and PL spectra measured on a pure F8BT thin film.

**Figure 4 nanomaterials-12-01883-f004:**
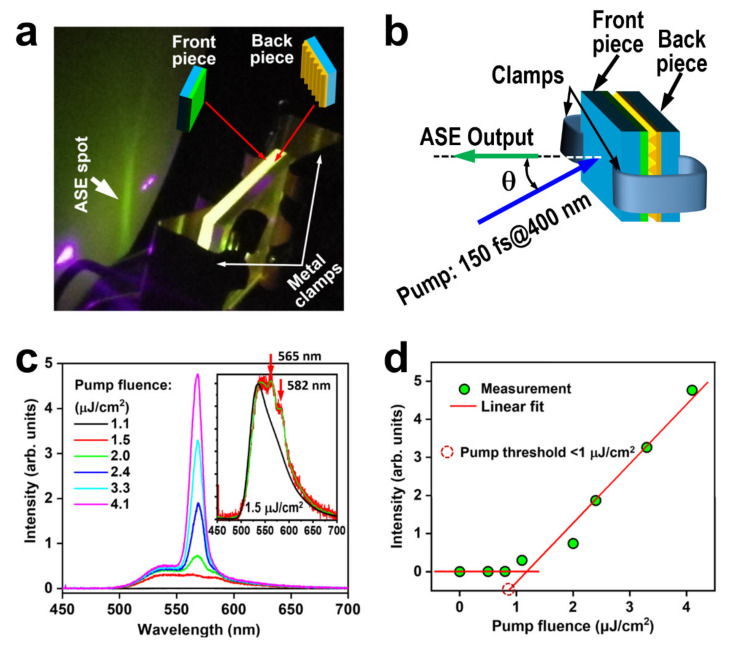
(**a**) Photograph of the experimental setup for characterizing the mechanically contact DFB microcavity and the ASE output generated in such an active microcavity. (**b**) Schematic illustration of the experimental configuration (θ≈20°). (**c**) ASE spectra at different pump fluences. Inset: comparison between the PL spectrum (black) and the emission spectrum (red) at a pump fluence of 1.5 μJ/cm^2^ with a guide curve (green). (**d**) Emission intensity as a function of the pump fluence and linear fittings to the measurement data, showing a center wavelength of ASE at 568.3 nm and a threshold pump fluence lower than 1 μJ/cm^2^.

## Data Availability

Not applicable.

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
