# Peer review of "Mechanically Contacted Distributed-Feedback Optical Microcavity"

_nanomaterials, 2022, doi:10.3390/nano12111883_

Round 1
Reviewer 1 Report
This manuscript reports novel design of amplification microcavity for amplified spontaneous emission. This study could contribute to new designs of microcavity LED or polymer lasers devices. The results are well organized and well described. Therefore, the manuscript could be published after following modifications.
- Authors mention that quality and surface smoothness and thickness-homogeneity of polymer microcavities in the abstract. However, they are not quantitatively indicated in the manuscript. They should be clearly indicated in the manuscript with evidence.
- The concept of amplification in microcavities are explained and described in Fig. 1a 4. However, the quality of picture is poor and it’s difficult to understand. The novel design concept should be more clearly indicated in the picture. Also, I think it is important to know the refractive index of F8BT, FS and PR. They should be shown in the manuscript.
- F8BT layer and PR grating is contacted with metal clamp. We couldn’t image how much is the strength of clamping. Please explain it in the manuscript. I think it is important parameter to achieve the proper contacting condition.
Followings are minor corrections
- The experimental condition is unclear in Fig. 3a. Please add schematic picture o experimental configuration.
- P3L100, “in the” is repeated.
Reviewer 2 Report
The authors report the construction of distributed feedback (DFB) optical microcavities. They are realized using mechanical contact between high-quality planar thin-film polymerics. Though the technique is interesting, several important performance parameters are missing. For example, what are the Q and losses of the cavity? How reproducible is this technique given the mechanical deformation during the attachment process? Also, what is the quantum efficiency of device?
Round 2
Reviewer 1 Report
I think the manuscript has been sufficiently improved to warrant publication in the journal.
Reviewer 2 Report
I would like to thank the authors for their response. The paper is suitable for publication.